# Quantitative expression of estrogen, progesterone and human epidermal growth factor receptor-2 and their correlation with immunohistochemistry in breast cancer at Uganda Cancer Institute

Henry Wannume[1,2]*, Nixon Niyonzima[1], Sam Kalungi[1,3], Julius Boniface Okuni[4], Tonny Okecha[1], Edward Kakungulu[1], Steven Mpungu Kiwuwa[5], Geoffrey Waiswa[1], Sylvester Kadhumbula[1], Monica Namayanja[6], Martin Nabwana[7], Jackson Orem[1]

1 Department of Clinical Support Services, Division of Laboratory and Pathology Medicine, Uganda Cancer Institute, Kampala, Uganda, 2 Department of Biomolecular Resources and Biolab Sciences, College of Veterinary Medicine, Animal Resources and Biosecurity, Makerere University, Kampala, Uganda, 3 Department of Pathology, Mulago National Referral Hospital, Kampala, Uganda, 4 Department of Veterinary Pharmacy, Clinical and Comparative Medicine, School of Veterinary Medicine and Animal Resources, Makerere University, Kampala, Uganda, 5 Department of Biochemistry, College of Health Sciences, Makerere University, Kampala, Uganda, 6 Centre for Biosecurity and Global Health Molecular Laboratory, Makerere University, Kampala, Uganda, 7 Division of Research, Makerere University—Johns Hopkins University Research Collaboration, Kampala, Uganda

* henry.wannume@uci.or.ug

**Data Availability Statement:** All relevant data are within the paper and its Supporting Information files.

## Abstract

The detection of Estrogen Receptor (ER), Progesterone Receptor (PR), and Human epidermal growth factor receptor 2 (HER-2) is important for the stratification of breast cancer and the selection of therapeutic modalities. This study aimed to determine the quantitative expression of ER, PR and HER-2 using Immunohistochemistry and their correlation with quantitative baseline Ct values measured using Quantitative Polymerase Chain Reaction (PCR). This study also assessed the use of fresh breast tissue biopsies preserved in RNAlater solution in the quantitative detection of these receptors using PCR technique. The study evaluated 20 matched formalin fixed paraffin embedded and RNAlater preserved samples for ER, PR, and HER-2 using IHC and quantitative PCR technique. One portion of the breast tissue biopsy was fixed immediately in 10% neutral buffered formalin and another was preserved in RNAlater. After the histological confirmation of breast cancer by the H&E technique, formalin fixed paraffin embedded tissues (FFPE)—positive cases were matched with their corresponding RNAlater samples for IHC and qPCR. The extracted RNA was quantified using Nanodrop technology, resulting into complementary DNA. ER and PR using IHC were expressed in 60% (n = 12) of the study samples and were negative in 40% (n = 8) of samples. HER-2 was negative in 70% (n = 14) of study samples, 25% (n = 5) positive, and 5% (n = 1) equivocal. With the quantitative expression of ER, PR, and HER-2 being reported in the IHC triple—negative breast cancer cases. The mean Ct values for the hormonal receptors correlated with what has been previously studied with ER at 19.631, PR at 25.410 and HER-2 at 25.695. There was no statistically significant difference between the mean Ct

**Funding:** This research was funded as part of the Master's Scholarship Study Support Award received from the African Development Bank under the Uganda Cancer Institute, Kampala, Uganda, received by Henry Wannume. Additional financial support for this research was awarded by The World Academy of Sciences (TWAS) (Grant No. 18-184 RG/BIO/AF/AC G FR3240303657) via a Master's student research support to Dr. Steven Mpungu Kiwuwa of the Department of Biochemistry, College of Health Sciences, Makerere University, Kampala, Uganda. This grant also supported Henry Wannume (Master's student) to cover part of the research fees.

**Competing interests:** The authors have declared that no competing interets exist.

values of RNAlater and FFPE with their P-values being 0.9919, 0.0896 and < 0.0001 for ER, PR, and HER-2 respectively. P-values; 0.9919 and 0.0896 for ER and PR respectively being greater than 0.05 it's a borderline significance although HER-2 had a statistical significance. With a concordance in the detection of these breast cancer hormonal receptors, qPCR can be used in our setting considering the delays that may be associated in following the samples through IHC processing.

## Introduction

Breast cancer is the most common cancer in women worldwide, accounting for 2.1 million new cancer cases in 2018 [1]. It is the second most common cancer in sub-Saharan Africa (SSA) and in Uganda, after cervical cancer [1]. The disease causes serious morbidity and mortality in Ugandan women with incidence and mortality rates of 21.3 and 10.3 per 100,000 population, respectively [2]. This indicates that about one-half of the Ugandan women diagnosed with breast cancer end up dying of the disease within the first five years of diagnosis. Although, breast cancer can be treated if diagnosed early and properly characterized [3], high mortality rates are attributed to late breast cancer diagnosis in women at stages III or IV which are very difficult to treat. Expression of the hormonal receptors for Estrogen (ER) and Progesterone (PR) as well as the Human epidermal growth factor receptor 2 (HER-2) are used for risk stratification and treatment selection for patients with breast cancer. There have been inconsistent results obtained in the determination of the expression of these receptors in SSA [4]. These reported inconsistencies in obtaining accurate and timely receptor results translate into delays in subjecting patients to appropriate therapy. This, in turn, results into progression of breast cancer into aggressive cancer types. Before the advent of immunohistochemistry (IHC) and molecular diagnostic techniques, risk stratification for breast cancer was not routinely done. Hence, the diagnosis of breast cancer was entirely based on histopathological tissue analysis using Hematoxylin and Eosin (H&E) stain and risk stratification using IHC was not routinely done. With the advent of specific or targeted therapies, breast cancers are now routinely assessed for receptor expression status of ER, PR, and HER-2, using IHC as a technique of choice [5]. The Immunohistochemistry technique has been used for many years in the detection of these receptors. With a noted increase in the number of triple-negative breast cancer reported cases in Uganda and the Uganda Cancer Institute of 34% [6]. This prevalence is almost three times of what has been reported in other countries, TNBC standing at 12–17% [6] and 5–10% as reported by Alexandra *et al.*, of 2023 [7] which need to be investigated. Triple-negative breast cancers (TNBC), a group of tumour that are negative for ER, PR, and HER-2 receptors. This has increased interest at the clinical, biological, and epidemiological levels due to the aggressive behaviour of the tumour, poor prognosis, and present lack of targeted therapies [8]. A better understanding of pathological mechanisms of TNBC onset and progression, including the still unclear association with BRCA1 mutations, and the causes of phenotypic heterogeneity may allow improvement in planning prevention and designing novel individualized treatments for this breast cancer subgroup [9]. Additionally, IHC is expensive and time-consuming requiring significant resources as well as training to be able to perform and interpret the tests. Consequently, only a limited number of places in SSA have access to IHC for diagnosis. However, in many parts of Africa, polymerase chain reaction technology is available because of developments set in place for HIV/AIDS, TB, and more recently for COVID-19 testing. It has been demonstrated before, that PCR technology can be used to determine the quantitative expression of receptors in cancer [10]. In Uganda, the gold standard for

determination of receptor expression profiles in breast cancer at present is IHC. This study was therefore used to determine the quantitative expression of the breast cancer receptors using quantitative polymerase chain reaction in relation to IHC which is the current gold standard. This study gives a good platform to test the practicability of qPCR in the detection of ER, PR, and HER-2 breast cancer hormonal receptors in a setting like the Uganda Cancer Institute. This is because no such study has been conducted at the institute and more so having the hormonal receptors quantified in RNAlater samples can serve as an alternative to shorten the expected result turnaround time. The study was also important in establishing whether there is a difference between freshly processed tissues without formaldehyde treatment and tissues fixed in formalin but with a reduced cold ischaemic time. This study also gave a proposition as highlighted by the results that qPCR can also be used as an alternative in the determination of breast cancer hormonal receptor expression. With this, qPCR can be used as a cheaper alternative to IHC in the quantitative determination of these hormonal receptors. Even though there are variations in cost, it is estimated that the costs associated with qPCR are less than half of the costs associated with IHC in the detection of ER, PR, and HER2 [11]. This study evaluated a cheaper alternative to IHC in the quantitative receptor determination which can result into improvements in the management of breast cancer at the Uganda Cancer Institute. This is important in establishing quantitative expression baseline data on the use of ER, PR, and HER-2 receptors among the breast cancer patients seen at the Uganda Cancer Institute which is the cancer referral hospital in the country. With this study, we hypothesized that if PCR measurement of receptor expression is comparable to IHC, it will provide a cheaper and more accessible method for the determination of breast cancer receptor expression. This could lead to better risk stratification and treatment selection for women with breast cancer in Uganda which will in turn lead to improved treatment outcomes and survival. This study also analyzed the possibility of using RNAlater preserved samples with the exclusion of buffered formalin in the quantitative determination of ER, PR, and HER-2 breast cancer receptors under qPCR technique.

## Materials and methods

### Study subjects

This was a cross-sectional study that was conducted among patients undergoing evaluation for breast cancer at the Uganda Cancer Institute. The study included all participants who consented for their breast tissue biopsies to be used in the study. The participants were biopsied after presenting with suspicious lesions on the mammography scans. The core biopsies were preserved and later used for breast cancer diagnosis using H&E and IHC ready-to-use antibody clones staining techniques. Relative quantitative expression of the hormonal receptors under study was also performed. Out of the twenty-four (24) participants who were screened for breast cancer, 20 participants were included in this study with these twenty (20) participants having confirmed breast cancer based on H&E staining technique. Gene glyceraldehyde-3 -phosphate dehydrogenase gene (GAPDH) was used as a housekeeping gene to provide a baseline reference point for each of the samples and used in the relative quantification expression experiments for the breast cancer hormonal receptor studies.

### Minimum sample size estimation

Based on the study by Sinn *et al.*, 2017 [12], where the correlation coefficient was 0.85 between qPCR and IHC, sample size was calculated using the formula for sample size correlation [13].

$$N = \left[\frac{Z_\alpha + Z_\beta}{C}\right]^2 + 3$$

$C = 0.5 \text{xIn} \left[\frac{1+r}{1-r}\right]$. From this N = 7.97, this being approximately 8 participants.

Where,

N = Required number of participants

α (two-tailed) = 0.05 = Threshold probability for rejecting the null hypothesis. Type I error rate.

β = 0.20 = Probability of failing to reject the null hypothesis under the alternative hypothesis. Type II error rate.

r = 0.85 = the expected correlation coefficient by [12]

The standard normal deviate for α = $Z_\alpha$ = 1.96

The standard normal deviate for β = $Z_\beta$ = 0.84

C = 0.5 * ln [(1+r)/ (1-r)] = 1.25615281

Since the sample size is 8, this study then considered a sample size of 20 participants as sufficient for this correlation experiment considering the available funds to run the IHC and qPCR tests on all the tissues.

The samples under study included 20 breast cancer participants who provided two portions of breast tissue biopsies each (one portion was preserved in RNAlater and the other FFPE tissues). All study participants provided written informed consent and the study was approved by the Uganda Cancer Institute Research Ethics Committee (approval number UCIREC REF-02-2020).

## Sample collection and sampling procedure of breast tissue biopsies

Written informed consent were obtained from all the study participants before their samples were used in the study and biopsies were performed under aseptic procedures by surgeons. Participant identifiers (study numbers or Lab IDs) were used in this study and the forms that were used for capturing data had these unique identifiers for easy tracking of the study samples. A total of twenty-four (24) participants were screened from which twenty (20) participants were selected for their tissue biopsies to be used in the study. Four (04) of the participants were negative for breast cancer under H&E hence excluded from this study and referred back to the breast clinic for guidance on the follow-up plan in line with early breast cancer diagnosis. Two sets of biopsies were picked. One of the collected breast tissue core biopsies was immediately fixed in 10% neutral buffered formalin solution for 24 hours and another small piece of the tissue kept in *RNAlater* (RNA protect Tissue Reagent, Qiagen, Germany, Catalogue number 76104). After an overnight storage at 4˚C, the *RNAlater* samples were then transferred to -20˚C. The samples preserved in 10% buffered formalin underwent a series of routine tissue processing steps and formalin fixed paraffin embedded (FFPE) tissue blocks were obtained from which H and E was performed. Then after the initial H&E histological confirmation of breast cancer using the FFPE blocks, the RNAlater samples were matched with the FFPE tissue blocks to aid in IHC and qPCR quantification says.

## RNA extraction and cDNA synthesis

Ribonucleic Acid (RNA) was extracted from the 20 matched FFPE and *RNAlater* preserved tissue biopsies that had been stored at -20˚C and thereafter cDNA synthesis was performed.

## Deparaffinization of FFPE tissues and RNA extraction

Ribonucleic acid (RNA) was extracted from the FFPE tissue samples, using RNeasy FFPE K.5421it Cat # 73504, 50T QIAGEN from Germany. The FFPE tissue sections used in this

study were 4 sections of 10-μm-thick obtained after confirmation of breast cancer tumour. The FFPE sections were deparaffinized by soaking them in 100% xylene with occasional shaking and vigorous vortexing for 10 seconds, at 10,000g and then incubated at 56˚C for 3 minutes. The tubes with the samples were cooled at room temperature. Proteinase K digestion (PKD) buffer of 240μL was added to the samples, vortexed and centrifuged at 11,000g for 1 minute. Ten microliters of Proteinase K was then added to the lower colourless phase and incubated at 56˚C and at 80˚C each for 15 minutes incubation with 5 minutes vortexing intervals. The lower colourless part was incubated on ice for 3 minutes and then centrifuged at 20,000g for 15 minutes. Twenty-five microliters of DNase booster buffer and 10μL of DNase 1 stock solution was added to the supernatant, vortexed and incubated at room temperature for 15minutes. Then added 500μL of RBC buffer to the sample to adjust the binding conditions. Seven hundred microliters of the sample mixture were transferred to the RNeasy MinElute and spun the column onto a new collection 2ml tube at 8,000g, this step was repeated until when the entire sample was used. Then added 500μL of RPE buffer to the RNeasy MinElute spin column and centrifuged at 8,000g for 15 seconds, this was repeated and centrifuged for 2 minutes. The RNeasy MinElute spin column was transferred onto a new 2ml tube, opened the lid and centrifuged for 5 minutes. Thirty microliters of RNase free water was used to elute the RNA in a 1.5 ml collection tube, centrifuged at 20,000g for 1 minute, quantified using a Nano-Drop and frozen at -80˚C until when cDNA was synthesized [14].

## Extraction of RNA from *RNAlater* persevered breast tissue biopsies

Ribonucleic Acid (RNA) from 0.15g to 1g weighed *RNAlater* stabilized breast tissue samples corresponding to the FFPE tissues was extracted using *RNeasy* Maxi kit (Cat # 75162) *from* Qiagen, Germany. The weighed tissues were transferred into different labelled 50ml falcon tubes with 1 ml RLT buffer (depending on the tissue weight) containing 10μL of β-mercaptoethanol and homogenized using tissue homogenizer 850 at 3000g. To limit any RNases during tissue homogenization, the tissue homogenizer probe was first cleaned using chloroform followed by 70% ethanol and autoclaved water. And this process was repeated every after homogenizing as sample to avoid cross contamination. The disrupted and homogenized tissues in appropriate volume of RLT buffer were then centrifuged at 4,000g for 10 minutes. Onto the supernatant in a 50ml falcon tube added 1 volume of 70% ethanol to the lysate and mixed vigorously. Transferred a minimum of 15ml of the sample to an RNeasy Maxi column onto a 50 ml tube and centrifuged at 4,000g for 5 minutes. This was followed by 15 ml of buffer RW1 and centrifuged at 4,000g for 5 minutes while discarding the flow-through liquids after each step of centrifugation. Then added 10 ml of buffer RPE to the column, centrifuged at 4,000g for 2 minutes, followed by another 10 ml of buffer RPE and centrifuged at 4,000g for 10 minutes to dry the RNeasy silica membrane. Ribonucleic acid was eluted into a new 50 ml falcon tube using 800 μL of RNase free water directly to the spin column membrane and left to stand for 1 minute. This was centrifuged at 4,000g for 3 minutes. Repeated the elution with another 800 μL and centrifugation at 4,000g for 3 minutes. Quantification of the eluted RNA was done using a NanoDrop and then stored at -80˚C until when cDNA synthesis was performed [14,15]. All the steps that were involved in the preparation and handling of total RNA were conducted in a biosafety class II cabinet under RNase free conditions.

## Complementary DNA (cDNA) synthesis

Complementary DNA (cDNA) was synthesized using a LunaScript RT SuperMix Kit (E3010S) (New England BioLabs, Hitchin, England) according to the manufacturer's instructions. A 20μL reaction mix comprising of 10μL (Up to 1 μg) of the RNA template, 4μL (1x) of

**Table 1. Primer sequences used with the z480 Roche system.**

| Receptor | Forward and Reverse Primer sequences | Product size (bp) | Reference |
|---|---|---|---|
| ESR1 (Estrogen receptor) | Forward primer: 5'-TCTGCAGGGAGAGGAGTTT-3'<br>Reverse primer: 5'-GGTCCTTCTCTTCCAGAGACTT-3' | 104 | [16]. |
| PGR (Progesterone receptor) | Forward primer: 5'-TCGAGTCATTACCTCAGAAGAT-3'<br>Reverse primer: 5'-CCCACAGGTAAGGACACCATA-3' | 80 | [16]. |
| ERBB2 (HER-2 receptor) | Forward primer: 5'-CAGCCCTGGTCACCTACAA-3'<br>Reverse primer: 5'-GGGACAGGCAGTCACACA-3' | 95 | [16]. |
| GAPDH | Forward primer: 5'-AAATCAAGTGGGGCGATGCTG-3'<br>Reverse primer: 5'-GCAGAGATGATGACCCTTTTG-3' | 127 | [17]. |

LunaScript supermix and 6μL of Nuclease free water was mixed in a PCR reaction tubes. The mixture was then incubated in a thermo cycler at temperatures of 25˚C, for 2 minutes to allow primer annealing, and 55˚C for 10 minutes to allow cDNA synthesis and the reaction was heat inactivated at 95˚C for 1 minute. The synthesized cDNA was then stored at -20˚C until qPCR tests were performed.

## Quantitative PCR (qPCR) amplification of estrogen, progesterone and human epidermal growth factor receptor 2 hormonal receptors

The ER, PR, HER-2 and GAPDH primers were procured from Eurofins Genomics Germany GmbH; Anzinger Str. 7a85560 Ebersberg Germany "Table 1" And the Luna Universal qPCR master mix was purchased from the New England BioLabs, (Hitchin, England). The qPCR reaction comprised of 2μL of cDNA (< 100ng) template, 1.5 μL (0.25 μM) of each of the primers, 10 μL (1x) of Luna Universal qPCR mix buffer, and 5 μL of Nuclease- free water in a final reaction volume of 20 μL. The qPCR reaction was performed in z480 Light cycler (Roche). The cycles involved; initial PCR denaturation at 95˚C for 60 seconds (one cycle), denaturation at 95˚C for 15 seconds, Annealing at 60˚C for 30 seconds and extension at 72˚C for 30 seconds for 45 cycles. A quantification step of one cycle was added at the end of the extension with a final extension at 72˚C for 10 minutes. The cycling conditions and the number of cycles for GAPDH was the same as for the hormonal receptor expressions under study. With GAPDH being used as a house keeping gene to generate baseline/background values for each of the individual samples under study. The reactions were set in duplicates and then the mean cycle threshold values (Ct) or Cp values were used to determine the relative amount of qPCR product by using GAPDH as a housekeeping gene which was run along all the samples under study. Therefore, GAPDH Ct /Cp values were used to determine the relative quantification of the receptors under study hence determining the over expression of estrogen, progesterone and Human epidermal growth factor receptor 2 in breast cancer.

## Accuracy in the amplified PCR products

In order to ensure accuracy in the RNA extraction and cDNA synthesis protocols, known IHC positive samples for ER, PR and HER-2 were selected, RNA extracted and their cDNA synthesized. Then with specific primers and Luna Universal qPCR Master Mix (M3003) and optimized protocol, a PCR was performed to detect probable amplification or non-amplification. Agarose gel 2%, was prepared and gel electrophoresis was performed in order to confirm the size of the base pairs (bp) for the PCR products obtained. Gene glyceraldehyde-3 -phosphate dehydrogenase gene (GAPDH) as a house keeping gene was also included in this experiment for assay normalization. The 2% agarose was prepared by using 2g of agarose powder into 100mls of Tris base, acetic acid and EDTA (1x TAE) in a microwavable flask. Let the mixture

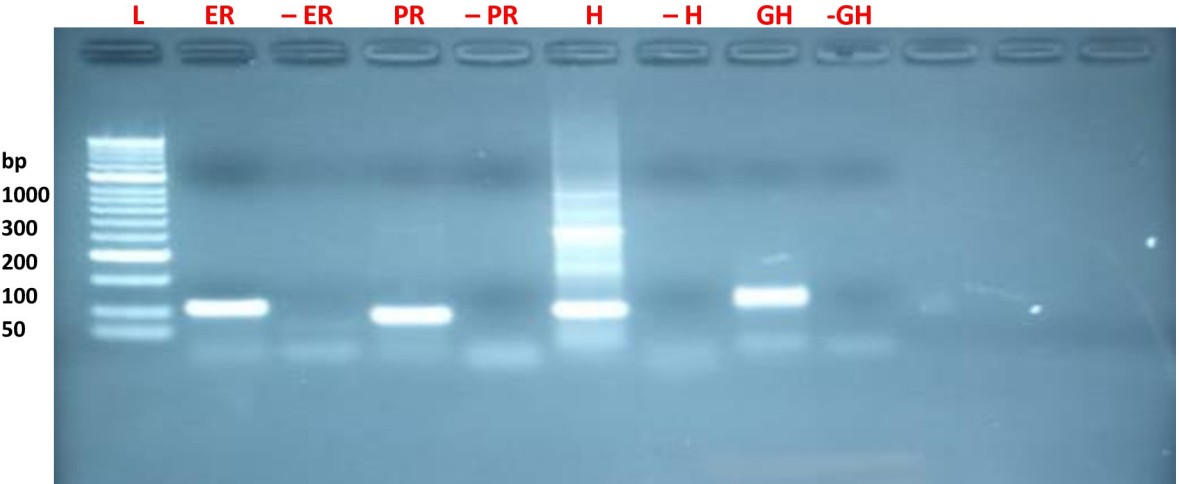

A 2% Agarose gel where L is 50bp ladder (bioline), ER is ESR1 positive, -ER is Negative, PR is PGR positive, -PR is PGR Negative, H is ERBB2 positive, -H is ERBB2 Negative, GH is GAPDH present and -GH is GAPDH absent. The expected band size for ESR1 is 104bp, PGR is 80bp, ERBB2 is 95bp and GAPDH is 127bp

**Fig 1. Two percent (2%) agarose gel indicating the amplified PCR products for ER, PR, HER-2 receptors and GAPDH.**

to boil for 2 minutes with 30 seconds mixing intervals. Left the agarose to cool at 56°C and then added 2μL (0.2 μg/ml) of ethidium bromide to aid in the visualization of the DNA bands. Then poured the agarose into a gel tray with the well comb in place and allowed set at room temperature for 45 minutes. Thereafter, loaded the samples using 1μL of loading dye and 5μL of the cDNA into their corresponding wells on the gel placed in an electrophoretic tank having 1x TAE buffer. With a 50 bp ladder in lane 1 to serve as a sizing measure. Gel electrophoresis was run using voltage of 150v, and 300Ma for 25 minutes. At the end of the run, visualized the gel in a gel imager for the DNA bands and captured the representative images "Fig 1".

## Normalization of the experiments

To calculate the normalized amount of the target, (Relative quantity, Rq, /over expression levels) the amount of the target was divided by the amount of the GAPDH (housekeeping gene gene). Given that the qPCR reactions were performed in duplicate, an average of the two was used and the GAPDH was performed alongside the other hormonal receptors for all the samples under study. The housekeeping gene hence was helpful in detection of the over-expression of ESR1, PGR and ERBB2 hormonal receptors in breast cancer. This study classified positives and negative cases of ER, PR and HER-2 based on the calculated over expression levels according to the cut-offs that had been studied by Manoj and others of 2020. Therefore, samples were noted to be positive under qPCR if their RQ /expression levels were ≥0.085, ≥0.0019 or ≥0.36 for ER, PR and HER-2 respectively [11].The concentration of the biomarker of interest (ER, PR or HER-2) was determined based on how soon the amplification curve crosses the threshold. A detectable signal was recorded in terms of fluorescence and plotted as a curve, this was the basis on which the equipment software calculates the Ct values. All samples that showed no detectable signals, their lines of expression remained below the threshold level "Fig 2".

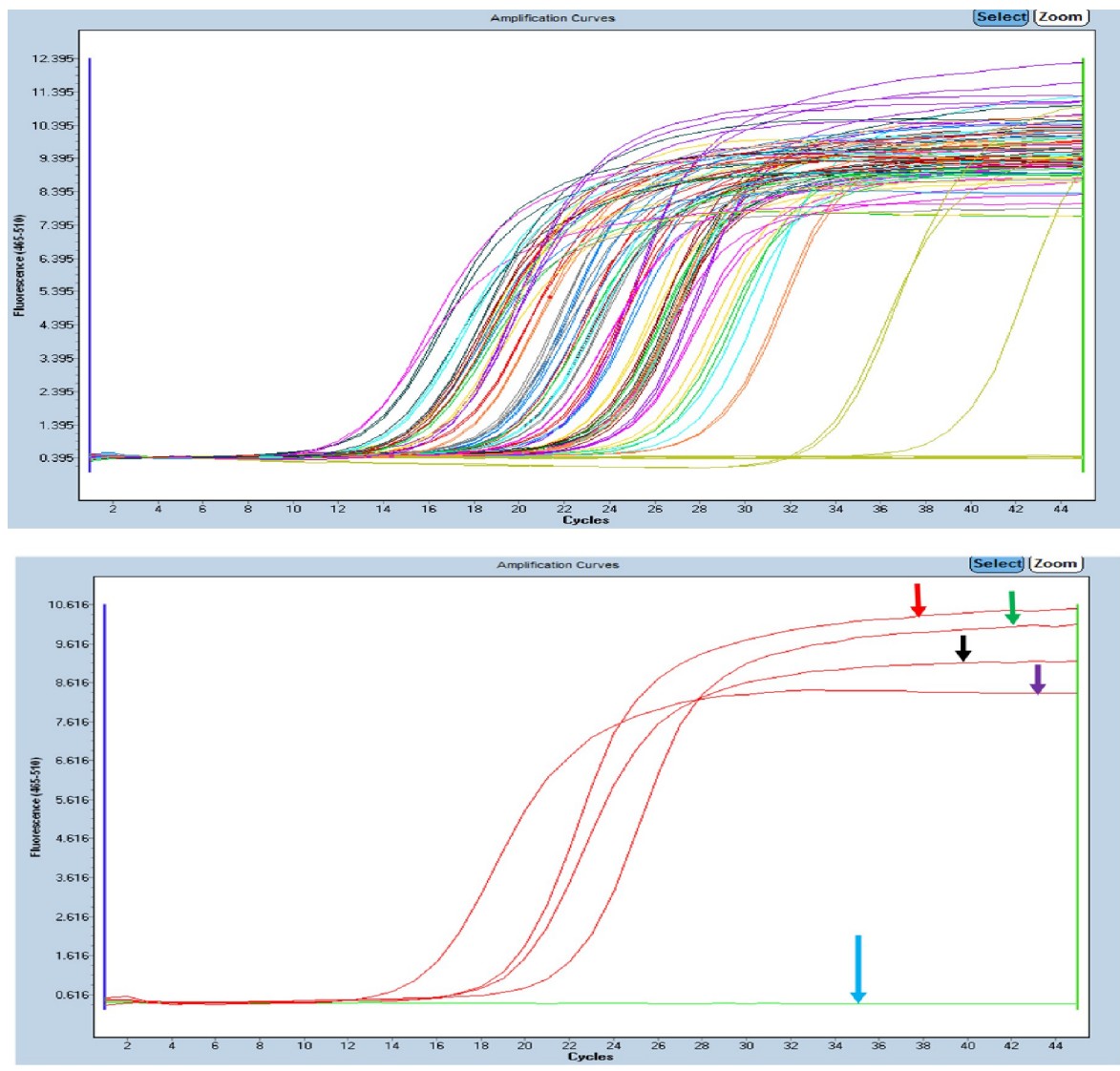

**Fig 2. Amplification plots for the quantitative expression of ER, PR, HER-2 and GAPDH.**

## Tissue processing for immunohistochemistry

Before IHCs were performed, the tissue blocks were selected after confirming the cancer diagnosis in such tissues under H&E staining. Sectioned 4μm of tissue and floated out the sections onto a charged slide labelled with the sample ID and the required receptor to be stained. Positive controls were also sectioned and stained along with the participant tissues. Staining was performed using Ventana Benchmark XT (Ventana medical systems, Inc. 1910 E. Innovation Park Drive, Tucson, AZ 85755, USA) and ready to use monoclonal antibodies. The rabbit monoclonal primary antibody from Ventana USA (anti-HER-2/neu (4B5), Confirm anti-Progesterone Receptor (PR) (1E2), Confirm anti-Estrogen Receptor (ER) (SP1) were used in the Immunostaining of the specific monoclonal antibody staining [18]. Interpretation Guide for VENTANA anti-HER2/neu (4B5). The labelled charged slides with 4μm of the tissue were placed in an oven set at 60°C for 30 minutes with HER-2 slides at room temperature for an overnight. This step was helpful in sticking of the tissue sections onto the charged slides in

addition to the adhesive nature of the slides that were used. Then loaded the reagents on the Benchmark XT equipment and allowed them to attain room temperature with slide barcode labels being applied. Loaded the individual slides with their corresponding ER, PR and HER-2 antibodies onto the ventana loading tray and started the equipment to perform the staining process. After the auto-staining process, the slides were immersed into soapy water to remove the liquid coverslip then transferred the slides into increasing concentrations of alcohol and cleared in xylene. The slides were then mounted with DPX, allowed the slides to air dry before microscopy was performed.

## Data analysis and interpretation

The equipment software was used to calculate the participant Ct values. The calculated Ct values and amplification plots were then downloaded from the results portal onto the computer and results recorded onto the worksheet template. Given that the IHC and qPCR data was obtained in different formats (scores based on staining intensity of the tumour cells, percentages and Ct values for qPCR), this data was then categorized in a form that could easily be compared. A semi-quantitative data set was generated based on the listed categories below in order to define the positive and negative cases for the two techniques and how high or low the cases were in relation to the studied hormonal receptors. For the qPCR data, the Ct values obtained were categorized as NA: Undetectable (0), Negative, Ct of ≥45 being undetectable (0): Negative, Ct 41–45: Low expression: (1+): Positive, Ct 31–40: Moderate expression: (2+): Positive and Ct 12–30: High expression levels: (3+): Positive [19].

However, for IHC data, reports on the ER, PR and HER-2 were given based on the whether the tumour cells took up the antibody of interest or not (staining intensity). For ER and PR, samples were considered to be positive or negative as indicated here; Negative if No staining (0) or 0% of nuclear staining of the tumour cells was noted. Less than 10% nuclear staining of the tumour cells (borderline), 1+, nuclear staining of 10% to 75% tumour cells, 2+ (positive) and a 3+ if greater than 75% nuclear staining of the tumour cells was recorded as positive [20]. With HER-2 being a membranous stain, the reporting tends to differ, 0: No staining is observed/membrane staining is in less than 10% of tumour cells (Score 0), Negative, 1+: if there is incomplete membrane staining that is faint or barely noticeable and with >10% of the invasive tumour cells; Negative. 2+ was reported once a weak to moderate complete membrane staining was observed in > 10% of tumour cells; Equivocal. And a 3+ once the circumferential membrane staining that is complete, intense and in >10% of tumour cells; Positive [21]. The cleaned data set was then exported to Stata Version 17.0 for correlation analysis. This data was then used to generate tables, distribution plots, graphs, histograms and a two t-test to under standard the correlation levels between two sample types. Comparison of qPCR and IHC results was done using a 3x3 tables to analyze the semi-quantitative categories of data obtained from each of the two of techniques. Finally, data that was obtained in form of Ct values for RNAlater and FFPE tissues was compared by note the mean differences between the two samples, the standard errors, range of the Ct spread and graphically by use of a histogram.

## Data quality control

The 10% buffered formalin that was used in fixation of tissues which were later used for IHC was freshly prepared prior to sampling and tissues were fixed immediately in formalin. During staining of IHC slides, known positive samples were stained to check the quality and specificity of the antibodies before study samples were processed. The stained slides were reported by a competent and certified pathologist with reviews being made by a consultant pathologist before reports were generated.

For qPCR, protocols were first optimized by setting convectional PCRs under different conditions, reactions and concentrations, then gel electrophoresis was performed on the PCR products to confirm the sizes or base pairs for the receptors under study. The qPCR assays were performed in duplicate, with GAPDH as a housekeeping gene being used for the normalization of these experiments. All this was being done with reviews made by the Molecular biologist with vast experience in relative quantitative PCR experiments. Known positive cases with high semi-quantitative rates under IHC for ER, PR and HER-2 were used as positives to check on amplification under the set conditions. Then a 2% agarose gel was set to run for sizing prior to qPCR of the other study samples. RNase free water was used as a negative control for these PCR experiments. The Ventana Bench Mark XT and Cobas 4800 equipment which were used in this study had been regularly maintained with updated service records.

## Results

### Demographic characteristics of participants

Of the twenty (20) participants studied, 95% (n = 19) of the participants were female with 5% (01) male being diagnosed with breast cancer. The median age of the participants was 41 years, with the youngest participant being 26 years and the oldest being 72 years. Four (04) participants were in the age range of 20–30 years, six (06) participants being between 31–40 years, two (02) participants were between 41–50 years, four (04) participants being between 51–60 years, two (02) participants were between 61–70 years and two (02) of the participants were aged between 71–80 years.

### Histological classification

Breast cancer tissues were screened and selected based on their morphological characteristics by reviewing the H&E stained slides under a microscope. Based on H & E histological diagnosis of breast cancer in the study participants, two morphological classifications of breast cancer were reported. Of the twenty (20) studied participants, 80% (18) were diagnosed with Ductal carcinoma of the breast with 20% (02) being Lobular carcinoma.

### Quantitative expression of ER, PR and HER-2 in breast cancer tissues

The quantitative expression of ER, PR and HER-2 based on the qPCR technique on the z480 light cycler was shown as amplification plots generated "Fig 2".

On a semi quantitative scale, the obtained Ct values were categorized to give a concentration strength of the of ER, PR and HER-2 biomarkers under study. Where NA was noted to be: Undetectable (0), Negative, Ct of ≥45 being undetectable (0): Negative, Ct 41–45: Low expression: (1+): Positive, Ct 31–40: Moderate expression: (2+): Positive and Ct 12–30: High expression levels: (3+): Positive.

The quantitative expression of ER showed the mean Ct value of 19.631, standard deviation (SD) of 15.259 and a median of 26.935. Basing on the semi-quantitative data presentation the ER Ct value mean was a strong expression in the category of 3+. ER receptor had a mean relative quantification of 0.682 with SD of 0.530 and a range of over expression level being 0–1.472 and a median value of 0.942.

Progesterone receptor (PR) gave a mean Ct value of 25.410, SD of 13.362 and a median of 30.653. Semi-quantitatively, the mean Ct value for PR was a strong to moderate expression at (3+). However, the mean relative quantification of PR was 0.854 with SD of 0.451 with a mean over expression range being from 0–1.36 and a median of 1.026. For HER-2, the mean Ct value was 25.695 with SD of 6.656 and a median of 26.748, semi-quantitatively the mean Ct

**Table 2. Quantitative expression of ER, PR and HER-2 from breast cancer tissues studied at the Uganda Cancer Institute.**

| Characteristic | Mean | Standard deviation | Range | Median |
|---|---|---|---|---|
| **Formalin samples** | | | | |
| ER | 19.631 | 15.259 | 0–41.76 | 26.935 |
| PR | 25.410 | 13.362 | 0–38.59 | 30.653 |
| HER-2 | 25.695 | 6.656 | 0–32.67 | 26.748 |
| **Extracted RNA** | 42.220 | 69.782 | 4–308.1 | 18.600 |
| **Over-expression levels** | | | | |
| ER | 0.682 | 0.530 | 0–1.472 | 0.942 |
| PR | 0.854 | 0.451 | 0–1.36 | 1.026 |
| HER-2 | 0.847 | 0.221 | 0–1.139 | 0.886 |

Quantitative baseline data for ER, PR and HER-2 receptors indicating their deviations from the mean range of the obtained data and their median. This data can be used as a point of reference in relating the positivity levels of the breast cancer hormonal receptors.

value was strong to moderate at 3+. HER-2 mean was 0.847 with SD of 0.221 and a range of over expression level of 0–1.139 with a median of value of 0.886 "Table 2".

## Expression of ER, PR and HER-2 based on immunohistochemistry staining technique

Of the twenty (20) samples that stained under IHC for ER, eight (08) had no nuclear staining at 0 (Negative), one (01) had less than 10% nuclear staining at 1+, borderline, four (04) participants had moderate nuclear staining (10% - 75%) defined as positive at 2+. With ER having a strong nuclear staining of above 75% staining of the tumour cells at 3+ this being positive. PR had eight (08) participants with no nuclear staining of the tumour cells (Negative), with one (01) participant having less than 10% nuclear staining of the tumour cells at 1+ (borderline). Moderate to strong staining of 2+ (10% - 75%) positive staining of the tumour cells was detected in six (06) participants and five (05) participants under PR showed a very strong staining of the tumour cells at 3+ (Positive). While HER-2 staining reported fourteen (14) participants with no membrane staining of the tumour cells at score zero, Negative, one (01) equivocal case was reported at 2+ that showed a weak to moderate membrane staining in greater than 10% of the tumour cells. Strong circumferential membrane staining that is complete and intense at greater than 10% of the tumour cells with anti-HER-2 of 3+ (positive) was reported in five (05) participants "Table 3".

The summarized IHC results and the initial diagnosis of breast cancer was reported in support of pathologists for the morphological reporting of H&E and later the stained IHC slides. "Fig 3" gives a representation of H&E stained slide with visible darkly stained tumour cells and HER-2 stained IHC slides to indicate the expression of hormonal receptors "Fig 3".

## Comparison of quantitative PCR and immunohistochemistry to determine expression of ER, PR and HER-2

Table 3 shows a 3X3 cross matrix to compare the results of qPCR and IHC on the breast cancer tissues. Of the seven (07) participants who tested negative under qPCR for ER, four (04) stained negative under IHC, with three (03) having a strong anti-ER staining at 3+ for IHC. One (01) participant had a weak positive quantification level of 1+ for qPCR with one 1+ for ER under IHC. Also, qPCR had five (05) participants with moderate positive expression (2+) while three (03) of the participants stained negative under IHC, with one (01) showing

**Table 3. Three by three table indicating the comparison between IHC and qPCR results for the expression of breast cancer hormonal receptors in breast cancer patients.**

| Receptors | QPCR | IHC n (%) | | | |
|---|---|---|---|---|---|
| | | Negative | 1+ | 2+ | 3+ |
| ER | Negative (n = 7) | 4(57.14) | 0(0.00) | 0(0.00) | 3(42.86) |
| | 1+ (n = 1) | 1(100) | 0(0.00) | 0(0.00) | 0(0.00) |
| | 2+ (n = 5) | 3(60.00) | 1(20.00) | 0(0.00) | 1(20.00) |
| | 3+ (n = 7) | 0(0.00) | 0(0.00) | 4(57.14) | 3(42.86) |
| PR | Negative (n = 4) | 2(50.00) | 0(0.00) | 1(25.00) | 1(25.00) |
| | 1+ (n = 0) | 0(0.00) | 0(0.00) | 0(0.00) | 0(0.00) |
| | 2+ (n = 10) | 6(60.00) | 1(10.00) | 1(10.00) | 2(20.00) |
| | 3+ (n = 6) | 0(0.00) | 0(0.00) | 4(66.67) | 2(33.33) |
| HER2 | Negative (n = 1) | 1(100.00) | 0(0.00) | 0(0.00) | 0(0.00) |
| | 1+ (n = 0) | 0(0.00) | 0(0.00) | 0(0.00) | 0(0.00) |
| | 2+ (n = 2) | 2(100.00) | 0(0.00) | 0(0.00) | 0(0.00) |
| | 3+ (n = 17) | 11(64.71) | 0(0.00) | 1(5.88) | 5(29.41) |

The analytical sensitivity of ER, PR and HER-2 cDNA qPCR in breast cancer samples was determined by first categorizing the obtained Ct values and degrees of over expression. With relative quantity of $\geq 0.085$, $\geq 0.0019$ or $\geq 0.36$ for ER, PR and HER-2 respectively being considered positive under qPCR in relation to the IHC gold standard.

moderate nuclear staining of the tumour cells and one (01) participant showed a strong positive expression (3+) under IHC. PR under qPCR quantification had four (04) negatives with two (02) being truly negative under IHC, one (01) moderate to strong nuclear staining (2+) and one (01) participant had a strong nuclear staining of the tumour cells at 3+ for IHC. Among the ten (10) participants who showed moderate positive quantification under qPCR at (2+), six (06) staining negative for PR under IHC, one (01) stained weakly at 1+, one (01) moderate to strong nuclear staining of the tumour cells at 2+ and two (02) participants showed a strong nuclear staining of the tumour cells at 3+ under IHC. Considering the HER-2 receptor, the one (01) participant who was negative under qPCR at zero (0) quantification also resulted as negative under IHC staining. The two (02) participants which were reported as moderate to positive under qPCR at 2+ turned out to be negative under IHC anti HER-2 staining. High expression levels for HER-2 under qPCR was reported in seventeen (17) of the study participants with eleven (11) turning out to be negative under IHC, one (01) had moderate weak to moderate membrane staining at 2+. And five (05) of the seventeen positives under qPCR turned out to be positive under IHC staining with a complete circumferential membrane staining at 3+ "Table 3".

## Effect of tissue preservation method on quantitative expression of ER, PR and HER-2 using RNAlater and formalin fixed tissues in breast cancer patients at the Uganda Cancer Institute

When quantitative expression of the three receptors that tend to be over expressed in breast cancer were compared for tissues fixed in 10% buffered formalin and RNAlater preservative the following results were obtained. (RNAlater being a solution that slows down the degradation of RNA in the tissues of interest). Generally, a positive quantitative over expression of ER, PR and HER-2 receptors was recorded in all the twenty (20) samples that had been preserved in RNAlater solution at 3+ level. While 13 of the ER cases gave a positive quantitative

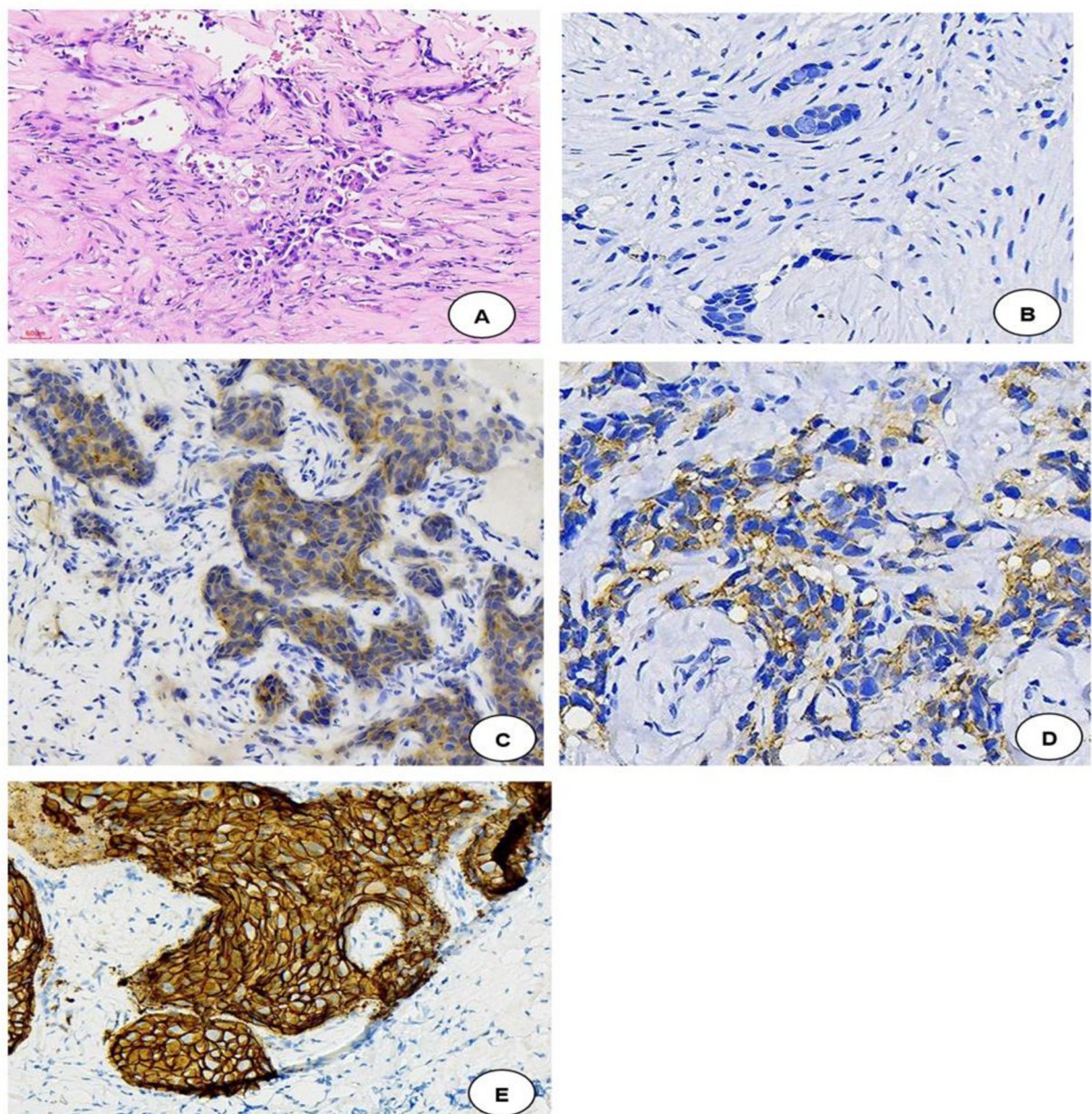

**Fig 3. Gives a representation of how H&E and IHC stained slides were graded/report.**

expression with FFPE samples, with 16 of the samples being positive and 19 of the cases being positive under HER-2 for the FFPE samples "Fig 4".

The quantitative expression of these hormonal receptors further provided a mean Ct valve for ER under RNAlater being 19.667 with SD of 3.372 while FFPE had mean Ct value of 19.631 with SD of 15.259. While the mean Ct value for PR in the two sample types was 19.927 and SD of 1.161 for RNAlater samples with a mean Ct value of 25.410 and SD of 13.362. Whereas, HER-2 had a mean Ct-value of 16.412 for RNAlater samples with SD of 2.4 with the FFPE samples having a mean Ct of 25.695 and SD of 6.695. This correlation between RNA later and FFPE breast tissue samples further provided the ER p-value 0.9919which was greater than

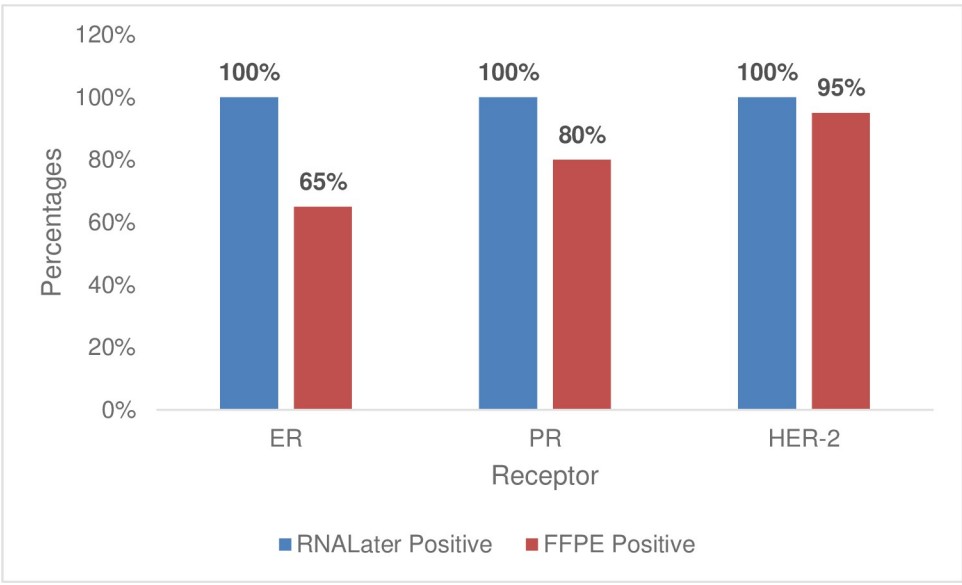

**Fig 4. Quantitative expression of ER, PR and HER-2 breast cancer receptors in RNAlater unfixed tissues and formalin fixed tissues (FFPE) using qPCR.**

0.05. However, there was a borderline significance between the Ct values of RNA later and FFPE breast tissue samples for the PR ($p = 0.0896$) and the FFPE Ct values were averagely higher than those of RNA later. Notably, there was a statistically significant mean difference between the Ct values of RNA later and FFPE breast tissue samples for the HER-2 receptor. The $p$-value ($<0.0001$) was less than 0.05 and the FFPE Ct values were also higher than those of the RNA later Ct values. Also, the degree of standard error was reported to be higher in the FFPE samples and compared to the unfixed RNAlater preserved samples "Table 4".

## Discussion

Despite Immunohistochemistry being the mostly used method in the detection of breast cancer hormonal receptors, it involves staining of ER, PR and HER-2 biomarkers in histological tissues. Although IHC is a reliable technique, it is also time-consuming due to the multiple

**Table 4. Quantitative expression of ER, PR and HER-2 from RNAlater and formalin fixed Breast cancer tissues using qPCR.**

| Characteristic | Mean (SD) | 95% CI | Difference, Mean (95%CI) | *p*-value |
|---|---|---|---|---|
| **ER** | | | | |
| RNAlater ER Ct value average | 19.667 (3.372) | 18.089, 21.244 | | |
| FFPE ER Ct value average | 19.631 (15.259) | 12.489, 26.772 | 0.036 (-7.302, 7.374) | 0.9919 |
| **PR** | | | | |
| RNAlater PR Ct value average | 19.927 (1.161) | 19.383, 20.470 | | |
| FFPE PR Ct value average | 25.410 (13.362) | 19.156, 31.663 | -5.483 (-11.899, 0.933) | 0.0896 |
| **HER-2** | | | | |
| RNAlater HER-2 Ct value average | 16.412 (2.400) | 15.292, 17.539 | | |
| FFPE HER-2 Ct value average | 25.695 (6.656) | 22.580, 28.810 | -9.280 (-12.508, -6.051) | **<0.0001** |

Mean differences, standard deviation error, 95% confidence interval and P-values for the quantitative expressions of ER, PR and HER-2 Breast cancer receptors using qPCR considering RNAlater (unfixed breast tissues) and FFPE samples.

sample processing steps each having lengthy waiting time periods and demands skilled pathologists. It is also worth reporting that most of the cancer subtypes are reported based on IHC due to the limited available datasets with gene expression-based subtyping [22], cancer margins in the diagnostic tissue, location of the tumour and tissue architecture of the tumour in tissue. Therefore, alternative methods like Quantitative PCR have to be thought of and this can be used to eliminate the need for expert cytologists, providing results which are independent of subjective interpretations a unique advantage of IHC [23]. Quantitative PCR also has well established protocols resulting into generation of reproducible results, highly sensitive and many samples can be processed or quantified within one testing hence confirming the expressed receptors [22]. However, qPCR cannot be used to determine where the markers are being expressed on the tissues sample i.e. whether in the tumour core or the margins which can easily be identified under IHC.

In this study, a combination of immunohistochemistry (IHC) and quantitative PCR methods of determining expression of Estrogen Receptors (ER), Progesterone Receptors (PR) and Human Epidermal Growth Factor 2 (HER-2) in Breast cancer tissues were compared. This involved recruiting participants from highly suspected individuals who had come to the Uganda Cancer Institute for breast cancer screening. Since no earlier quantitative expression of the breast cancer hormonal receptors had been reported at the pathology laboratory, the study was essential in having baseline data generated. This study established baseline data for the quantitative expression of ER, PR and HER-2 in breast cancer at the UCI which can be used as a reference for future studies. The obtained Ct values were directly proportional to the concentration of these breast cancer hormonal receptors which averages were used to calculate the degree of over expression using GAPDH as a housekeeping gene.

Having well established, reproducible and reliable mean Ct values for these key determinant biomarkers in the selection of breast cancer treatment makes it possible to have RT-PCR test used hand in hand with IHC. This study therefore reports mean Ct values which can later be used as baseline assessments of breast cancer tissues. The mean Ct value for ER was 19.631 while the PR it was 25.410 and mean Ct value for HER-2 was 25.69. These results were in agreement with mean Ct values for HER-2 that have been reported in positive HER-2 tumours at different concentration levels ranging from mean Ct values of RT-qPCR of 19.45, 23.20, 26.49, 29.79, 33.35, and 36.08 [24].The Results from this current study were also very closely similar to the Ct value cut-offs that have been studied for a positive expression of ER, PR and HER-2 hormonal receptors to be 22.18667, 18.2038, and 23.69193, for ER, PR and HER-2 respectively [25]. Though our study had a smaller sample size.

While comparing the over expression levels as per the study findings, this study reported mean expression levels of ER at 0.682, PR had 0.854 with HER-2 having a mean expression level of 0.847. These findings are very close to the findings of Iverson and others who reported ER expression cut off for IHC at 0.842 (95% CI: 0.693–0.992), with PR cut off at 0.861 (95% CI: 0.730–0.993) and HER-2 status being 0.776 (95% CI: 0.592–0.961) being reported [16]. However, our study had a slightly lower sample size. Results from this study also were closely correlating with the study findings on ER and PR amplification using qPCR and SYBER green as a dye of choice [26].

When qPCR and IHC data was correlated, about 30% (n = 6) of the participants samples studied under Immunohistochemistry had triple negative breast cancer which is close to other studies [26].Triple negative cancers are considered to be one of the most aggressive type of breast tumours with fewer targeted therapies. This hence makes it important to have such cancers stratified to better guide in the different breast cancer treatment modalities. When quantitative polymerase reaction (qPCR) was performed on RNAlater cDNA samples, all the TNBC samples showed amplification with an over expression level of close to 1.5 considering

GAPDH as housekeeping gene. This indicates that there was a notable quantitative amplification of the ER, PR and HER-2 for the IHC reported triple negatives. These findings hence imply that some of the triple negative hormonal receptors may not truly be negatives for such receptors. This confirms that there may be need to perform quantitative expression studies for such hormonal receptor negative cases. It should also be noted that the quantitative expression of these ER, PR and HER-2 hormonal receptor quantifications were significantly lower in IHC samples which had lower scores of (0, 1+ and 2+) compared to samples which were 3+ based on the reported results. This finding concurred with a previous study which highlighted HER2 amplification levels being significantly lower in tissue samples which had low IHC expressions (0 or 1+) than those in samples with high expression (3+) [27]. A critical analysis of the qPCR and IHC results resulted into some discordant results which cannot clearly be identified though some reports suggest the effect of formalin on the expression of the hormonal receptors under study. This may have resulted into lower expression levels or completely giving negative result out comes. Discordance in results between IHC and molecular based studies has been reported by previous studies [28].

While relating the mean Ct values obtained for ER, PR and HER-2 breast cancer hormonal receptors for samples fixed in formalin with Ct values for tissues not being fixed in formalin being lower within a positive range category compared to those in formalin "S1–S4 Tables". Therefore with P-values of 0.9919, 0.0896 and <0.0001 for ER, PR and HER-2 respectively between ER, PR and HER-2 indicated a borderline, moderate and significant variation in the Ct values obtained from RNAlater samples and the formalin fixed samples. This study also noted a difference in the concentration of RNA that was extracted from fresh unfixed tissues (RNAlater) and formalin fixed breast cancer tissues "S5 Table". An indication of a probable interference of formalin fixation on the expression of these breast cancer hormonal receptors. This difference in the positivity rate in the expression of these hormonal receptors between the two samples could be related to the formalin effect possibly by masking the expression of these hormonal receptors hence more negatives being realized. In a related study that was done to compare the fresh frozen tissues and formalin-fixed paraffin embedded tissues in gene mutation analysis using a multi-gene panel in colorectal cancer patients, interesting results were reported. Of the 129 gene variants that were studied, 96 variants were identified in both FFPE and fresh frozen tissues with 27 variants being found only in FFPE tissues and 6 variants found in fresh frozen tissues only. This concordance of more than 74% between these two samples types gives an indication that other sample types other than formalin can be used in studying molecular markers in tissues. These findings also indicate that formalin may not fully be masking the presentation of genes in formalin fixed samples since even extra genes were isolated in formalin fixed tissues more than frozen samples [29]. In order to determine whether there was no amplification in this relative quantitative experiment, reference was made to define the missing values (N/A) in quantitative experiment and low undetectable levels of expression hence no amplification [30]. Therefore, this comparison of qPCR results with the IHC semi-quantitative data showed that qPCR can be used to determine ER, PR and HER-2 hormonal receptor status within a shorter time and in a cost-effective way. However, a bigger sample size with different sample types is needed for more studies to better establish and define the cut-offs and degree of over expression using a housekeeping gene.

## Conclusions

Given the fact that qPCR managed to quantitatively determine the over expression levels of ER, PR and HER-2, qPCR can be used in our setting based on the established mean Ct values. The study gave a very good correlation between qPCR and IHC in the quantitative expression

of breast cancer receptors. However, there is need to understand why three (03) ER, and one (01) PR samples had a strong positive expression at 3+ under IHC yet they presented negative under qPCR.

The study also concludes that there is a possibility of formalin masking the ER, PR and HER-2 breast cancer receptors. This is because all formalin free samples (RNAlater preserved) showed 100% quantifications with only 65%, 80% and 95% quantitative expression for ER, PR and HER-2 respectively for formalin fixed samples. Hence Breast cancer tissue biopsies preserved in RNAlater can also be used in the quantitative determination of ER, PR and HER-2 receptors.

## Recommendations

Based on the obtained results from the study, we recommend the following;

Quantitative PCR (qPCR) need to be applied on all triple negative breast cancer cases diagnosed by IHC since not all hormonal receptor IHC negatives were all negative by qPCR. There is need to conduct a bigger study with more sample types like FNACs, fresh frozen tissue biopsies, RNAlater samples and formalin to continue to assess the formalin fixation effect on the masking of ER, PR and HER-2 receptors in breast cancer. Further studies including a larger number of samples need to be conducted in order to have the reported receptor Ct values checked at a larger scale to clearly define the cut-off values.

## Supporting information

**S1 File. Data.**
(XLSX)

**S1 Table. Ct values for FFPE cDNA samples (ER).**
(TIF)

**S2 Table. Ct values for RNAlater stabilized cDNA samples (PR).**
(TIF)

**S3 Table. Ct values for FFPE cDNA samples (PR).**
(TIF)

**S4 Table. Ct values and expression levels for RNAlater stabilized cDNA samples (HER-2).**
(TIF)

**S5 Table. Showing RNA concentrations extracted from RNAlater and FFPE samples.**
(TIF)

## Acknowledgments

We thank all the study participants for accepting to have their samples used in the study. We thank the study team Mr. Wasswa Hassan, Prossy Namuli, David Kasozi and the entire UCI Pathology Laboratory for all the support rendered to us during the conduct of this study. We thank Professor Carlos Caldas Breast cancer research group at the University of Cambridge, Cancer research UK, Dr. Suet- Feung Chin, Geoffrey Ssentamu and David Nalumenya for the technical support to this study.

## Author Contributions

**Conceptualization:** Henry Wannume.

**Data curation:** Henry Wannume, Martin Nabwana.

**Formal analysis:** Henry Wannume, Geoffrey Waiswa, Sylvester Kadhumbula, Monica Namayanja, Martin Nabwana.

**Funding acquisition:** Henry Wannume, Jackson Orem.

**Investigation:** Henry Wannume.

**Methodology:** Henry Wannume, Tonny Okecha, Edward Kakungulu, Geoffrey Waiswa, Monica Namayanja.

**Project administration:** Henry Wannume, Jackson Orem.

**Resources:** Henry Wannume, Steven Mpungu Kiwuwa.

**Software:** Henry Wannume, Martin Nabwana.

**Supervision:** Henry Wannume, Nixon Niyonzima, Sam Kalungi, Julius Boniface Okuni, Tonny Okecha, Edward Kakungulu, Steven Mpungu Kiwuwa, Monica Namayanja, Jackson Orem.

**Validation:** Henry Wannume, Nixon Niyonzima, Sam Kalungi, Julius Boniface Okuni, Tonny Okecha, Edward Kakungulu, Steven Mpungu Kiwuwa, Geoffrey Waiswa, Sylvester Kadhumbula, Monica Namayanja.

**Visualization:** Henry Wannume, Sam Kalungi, Julius Boniface Okuni, Tonny Okecha, Edward Kakungulu, Steven Mpungu Kiwuwa, Geoffrey Waiswa, Sylvester Kadhumbula, Monica Namayanja.

**Writing – original draft:** Henry Wannume.

**Writing – review & editing:** Henry Wannume, Nixon Niyonzima, Sam Kalungi, Julius Boniface Okuni, Tonny Okecha, Edward Kakungulu, Steven Mpungu Kiwuwa, Geoffrey Waiswa, Sylvester Kadhumbula, Monica Namayanja, Martin Nabwana, Jackson Orem.

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
