## [Decision Letter · Decision Letter 0]

28 May 2024

PONE-D-24-14363Original Manuscript

Title- Quantitative Expression of Estrogen, Progesterone and Human epidermal growth factor receptor-2 and their correlation with Immunohistochemistry in Breast Cancer at Uganda Cancer InstitutePLOS ONE

Dear Dr. Wannume,

Thank you for submitting your manuscript to PLOS ONE. After careful consideration, we feel that it has merit but does not fully meet PLOS ONE’s publication criteria as it currently stands. Therefore, we invite you to submit a revised version of the manuscript that addresses the points raised during the review process.

Thank you for submitting the manuscript. The limited number of samples is a big concern among the reviewers and me. Also, I think the manuscript contains a lot of redundancy. For example, the Method section tells a lot about qPCR, but this is a very fundamental method. I think that the author could shorten this basic point. Also, the Abstract should be shortened. For instance, the information about the housekeeping gene is not necessary for the abstract.

We want to re-evaluate after Major revision. Please refer to the reviewers' comments. We will look forward to resubmission.

We look forward to receiving your revised manuscript.

Kind regards,

Kenji Fujiwara, PhD, MD

Academic Editor

PLOS ONE

2. During your revisions, please note that a simple title correction is required: We request you to please remove the phrase "Original manuscript Title" from the title of your manuscript. Please ensure this is updated in the manuscript file and the online submission information.

5. We note that you have referenced (UCI. (2018). Statistical unpublished data Uganda Cancer Institute) which has currently not yet been accepted for publication. Please remove this from your References and amend this to state in the body of your manuscript: (ie “Bewick et al. [Unpublished]”) as detailed online in our guide for authors

7. Please remove your figures from within your manuscript file, leaving only the individual TIFF/EPS image files, uploaded separately. These will be automatically included in the reviewers’ PDF.

Additional Editor Comments:

Dear Dr. Wannume.

Thank you for submitting the manuscript. The limited number of samples is a big concern among the reviewers and me. Also, I think the manuscript contains a lot of redundancy. For example, the Method section tells a lot about qPCR, but this is a very fundamental method. I think that the author could shorten this basic point. Also, the Abstract should be shortened. For instance, the information about the housekeeping gene is not necessary for the abstract.

We want to re-evaluate after Major revision. Please refer to the reviewers' comments. We will look forward to resubmission.

Best regards,

Kenji Fujiwara

Reviewers' comments:

Reviewer's Responses to Questions

**Comments to the Author**

1. Is the manuscript technically sound, and do the data support the conclusions?

Reviewer #1: No

Reviewer #2: Yes

Reviewer #3: Yes

Reviewer #4: Yes

2. Has the statistical analysis been performed appropriately and rigorously? 

Reviewer #1: Yes

Reviewer #2: Yes

Reviewer #3: Yes

Reviewer #4: Yes

3. Have the authors made all data underlying the findings in their manuscript fully available?

Reviewer #1: Yes

Reviewer #2: No

Reviewer #3: Yes

Reviewer #4: Yes

4. Is the manuscript presented in an intelligible fashion and written in standard English?

Reviewer #1: Yes

Reviewer #2: Yes

Reviewer #3: Yes

Reviewer #4: Yes

5. Review Comments to the Author

Reviewer #1: The study correlates IHC and qPCR data in FFPE samples of breast cancer. The sample size of the study is small. Further, there are subgroups within the limited number of samples like 1+, 2+ and 3+. The argument that RT-PCR is cheaper than IHC seems unjustified.

Reviewer #2: • Authors have stated that data is fully available without restriction, but under the section describing where the data may be found there is a link to the author’s ORCiD account, not a data repository. Kindly correct this and add a link to the data or upload as a separate file.

• Please check if format of citations and modify according to journal criteria where needed.

• In Introduction, line number 77, please clarify the following “This is almost three times what is reported in other regions”.

• Minor English language proofreading by the authors will improve the manuscript further.

• The authors propose qPCR as a cheaper and faster method of determining hormonal receptors expression in breast cancer, as compared to IHC. However, in their discussion the authors should also mention the benefits associated with IHC which are missed when using qPCR, for example, it cannot be determined WHERE the markers are being expressed, whether in the tumour core, the margins etc, IHC also informs us regarding tissue architecture of tumour etc. Please mention benefits of both techniques for a more comprehensive discussion.

• It is not mentioned whether IHC antibodies were diluted for use, or these were ready-to-use clones.

Reviewer #3: Dear Author,

Thank you for your submission. It was interesting read. However, the study sample size of 20 is too small to draw any concrete conclusion and recommendations. Also, the reference are not marked (numbered) in the body of the manuscript as per the references. The manuscript is otherwise well written.

Best wishes,

Reviewer

Reviewer #4: The manuscript is well-written and presents an original contribution to the field. Your comprehensive methodology is well-detailed and robust, ensuring the reproducibility and reliability of your findings.

The results are compelling and well-supported by data. Overall, your research is a significant addition to the existing literature, and I believe it will be of great interest to the specific group

6. PLOS authors have the option to publish the peer review history of their article (what does this mean?). If published, this will include your full peer review and any attached files.

Reviewer #1: No

Reviewer #2: No

Reviewer #3: No

Reviewer #4: No

---

## [Author Response · Author response to Decision Letter 0]

24 Aug 2024

All the reviewers and editors' comments have been addressed well as per the attached letter labelled as "Response to Reviewers" with a table summarizing the concerns raised by the reviewers and editors with the specific responses made. This also indicates the specific corrections or improvements that have been made on the manuscript.

---

## [Decision Letter · Decision Letter 1]

16 Sep 2024

Title- Quantitative Expression of Estrogen, Progesterone and Human epidermal growth factor receptor-2 and their correlation with Immunohistochemistry in Breast Cancer at Uganda Cancer Institute

PONE-D-24-14363R1

Dear Dr. Wannume,

We’re pleased to inform you that your manuscript has been judged scientifically suitable for publication and will be formally accepted for publication once it meets all outstanding technical requirements.

Kind regards,

Kenji Fujiwara, PhD, MD

Academic Editor

PLOS ONE

Additional Editor Comments (optional):

Dear Dr. Wannume.

Thank you for revising your manuscript appropriately. I think in the manuscript you responded to our concerns well. We agreed to the acceptance.

Yours sincerely,

Kenji Fujiwara

Academic editor

Reviewers' comments:

Reviewer's Responses to Questions

**Comments to the Author**

1. If the authors have adequately addressed your comments raised in a previous round of review and you feel that this manuscript is now acceptable for publication, you may indicate that here to bypass the “Comments to the Author” section, enter your conflict of interest statement in the “Confidential to Editor” section, and submit your "Accept" recommendation.

Reviewer #2: All comments have been addressed

2. Is the manuscript technically sound, and do the data support the conclusions?

Reviewer #2: Yes

3. Has the statistical analysis been performed appropriately and rigorously? 

Reviewer #2: Yes

4. Have the authors made all data underlying the findings in their manuscript fully available?

Reviewer #2: Yes

5. Is the manuscript presented in an intelligible fashion and written in standard English?

Reviewer #2: Yes

6. Review Comments to the Author

Reviewer #2: Authors have taken care to respond satisfactorily to all of the reviewer's concerns.

Article includes a discussion of the pros and cons of both techniques. Also, authors have made their data available.

7. PLOS authors have the option to publish the peer review history of their article (what does this mean?). If published, this will include your full peer review and any attached files.

Reviewer #2: No

---

## [Editor Report · Acceptance letter]

30 Oct 2024

PONE-D-24-14363R1 

PLOS ONE

Dear Dr. Wannume, 

I'm pleased to inform you that your manuscript has been deemed suitable for publication in PLOS ONE. Congratulations! Your manuscript is now being handed over to our production team.

Kind regards, 

on behalf of

Dr. Kenji Fujiwara 

Academic Editor

PLOS ONE